# Clinical Significance of the Immunohistochemical Expression of Histone Deacetylases (HDACs)-2, -4, and -5 in Ovarian Adenocarcinomas

**DOI:** 10.3390/biomedicines12050947

**Published:** 2024-04-24

**Authors:** Georgia Levidou, Dimitrios Arsenakis, Dimitrios I. Bolovis, Roxanne Meyer, Cosima V. M. Brucker, Thomas Papadopoulos, Stamatios Theocharis

**Affiliations:** 1Department of Pathology, Medical School, Klinikum Nuremberg, Paracelsus University, 90419 Nuremberg, Germany; nicoleroxanne.meyer@klinikum-nuernberg.de (R.M.); thomas.papadopoulos@klinikum-nuernberg.de (T.P.); 2Department of Gynecology and Obstetrics, Medical School, Klinikum Nuremberg, Paracelsus University, 90419 Nuremberg, Germany; dimitrios.arsenakis@klinikum-nuernberg.de (D.A.); dimitrios.bolovis@klinikum-nuernberg.de (D.I.B.); cosima.brucker@klinikum-nuernberg.de (C.V.M.B.); 3First Department of Pathology, National and Kapodistrian University of Athen, 11527 Athens, Greece; stamtheo@med.uoa.gr

**Keywords:** HDAC, ovarial adenocarcinoma, immunohistochemistry, prognosis

## Abstract

Background: Histone deacetylases (HDACs) are implicated in carcinogenesis, and HDAC inhibitors (HDACis) are explored as a therapeutic tool in several tumors. The aim of this study was to evaluate the clinical significance of HDAC-2, -4, and -5 expression in epithelial ovarian carcinoma (EOC). Methods: HDAC-2, -4, and -5 immunohistochemical expression was examined in 92 EOC tissue specimens and was correlated with clinicopathological characteristics. Results: HDAC-2 was the most frequently (94.4%) expressed isoform, being marginally higher in serous tumors compared with other types (*p* = 0.08). HDAC-5 was the less frequently expressed (28.1%), being positively associated with HDAC-4. HDAC-4 positivity was associated with lower FIGO-stage (*p* = 0.045) and T-category (*p* = 0.043) and the absence of lymph node (*p* = 0.05) or distant metastasis (*p* = 0.09) in serous carcinomas. HDAC-2 positivity was correlated with the absence of lymph node metastasis in serous tumors (*p* = 0.045). On the contrary, HDAC-5 nuclear positivity was correlated with lymph node metastasis in the entire cohort (*p* = 0.048). HDAC-4 positivity was marginally associated with favorable prognosis in serous carcinomas in univariate survival analysis (*p* = 0.086), but this correlation was not significant in multivariate analysis. Conclusions: These findings suggest a differential expression among HDAC-2, -4, and -5 in ovarian adenocarcinomas in terms of immunolocalization, positivity rate, and associations with clinicopathological parameters, providing evidence for a potential role in the pathobiology of EOC.

## 1. Introduction

Epigenetic alterations are essential regulators of gene transcription and have been reportedly correlated with carcinogenesis and tumor progression. These post-translational modifications include acetylation, methylation, phosphorylation, ubiquitination, sumoylation, and ADP-ribosylation of the long N-terminal extensions of the DNA-bound core histones [1]. Acetylation is regulated by the balance of the following two opposite classes of enzymes: histone acetyl transferases (HATs), which transfer the acetyl moiety from acetyl coenzyme A to specific lysine residues of histones, and histone deacetylaces (HDACs), which are responsible for removing acetyl groups [2,3].

The members of the HDAC protein family are divided into four classes according to the corresponding homology to the yeast S. Cerevisiae [1,2,4]. Among them, class I HDACs (HDAC-1, -2, -3, and -8) show homology with yeast proteins Rpd3, Hos1, and Hos2 [5]. Class I and II members share a common enzymatic mechanism, the Zn-catalyzed hydrolysis of the acetyl–lysine amide bond [5]. Class II HDACs are divided into two different subclasses (IIa and IIb) and comprise HDAC-4, -5, -6, 7, -9, and 10, which share homology with yeast proteins HDA1 and Hos3 [5,6]. HDAC-4, -5, -7, and -9 belong to class IIa and seem to be associated with members of the MEF and Runx families. Class IIb encompasses HDAC-6 and -10. Class III HDACs (sirtuins) are structurally distinct from classes I and II HDACs [4]. Class IV presents with an additional Zn-dependent HDAC (HDAC-11) that is evolutionary different from HDAC classes I/II [1,4,5,6].

HDAC proteins have been shown to be expressed in a wide range of human malignant tumors, being associated with tumor initiation and disease progression as well as patients’ prognosis [6]. In various tumors, a higher level of HDACs is associated with advanced disease and poor prognosis. There are, however, multiple reports suggesting an important role of HDACs in the prevention of metastasis being associated with better clinical outcomes [7,8]. The mechanisms by which individual members of the HDAC protein family can regulate tumor development are quite diverse, and several reports suggest a distinct type-specific role of HDAC overexpression [9]. HDACs induce a range of cellular and molecular effects through the hyperacetylation of histone and nonhistone substrates and therefore could either repress tumor suppressor gene expression or regulate the oncogenic cell-signaling pathway via the modification of key molecules [10].

There is limited information available to date regarding the clinical significance of different classes of HDAC expression in ovarian adenocarcinomas and in particular class II members, even though the role of HDAC inhibitors in the treatment of ovarian adenocarcinoma is being thoroughly discussed [11]. The present study aims to assess the immunohistochemical expression of several HDACs retrospectively (classes I and IIa) in specimens of ovarian adenocarcinomas, in association with clinicopathological parameters as well as patient overall survival.

## 2. Materials and Methods

### 2.1. Patient Collective

This is a retrospective investigation of archival histological material from patients with ovarian carcinoma diagnosed from 2016 to 2018 at the Department of Pathology, in Klinikum Nuremberg, Germany. All the specimens with the diagnosis of epithelial ovarian cancer on the primary site and with enough material for further immunohistochemical analysis were included in the present investigation. Metastatic lesions or other histological types of malignant ovarian tumors were excluded. During this procedure, we collected 92 specimens. None of the patients included in the present study had received any kind of neoadjuvant radio- or chemotherapy prior to the surgical procedure. A clinical stage was assigned according to the International Federation of Obstetrics and Gynecology (FIGO) standards, based on surgical and pathological findings as well as postoperative abdominopelvic computerized tomography scans. This study was conducted in accordance with the Declaration of Helsinki and was approved by the internal institutional Bioethics Committee of Paracelsus University, Nuremberg, Germany. 

### 2.2. Immunohistochemical Analysis

Immunohistochemistry (IHC) for HDAC-2, -4, and -5 was performed on formalin-fixed, paraffin-embedded tissue sections. From each paraffin block, a 4 µm thick section was cut, mounted, and dried at 37 °C. Subsequently, sections were stained with hematoxylin and eosin using an Automatic Stainer (BenchMark Ultra, VentanaRoche; Roche Diagnostics GmbH, Germany), according to the manufacturer’s protocol and using the following antibodies: rabbit polyclonal anti-HDAC-2 (Y461, ab32117, Abcam, Cambridge, U.K.), mouse monoclonal anti-HDAC-4 (ab16339, Abcam, Cambridge, UK), and mouse monoclonal anti-HDAC-5 (ab55403, Abcam, Cambridge, UK). An OptikView DAB Detection kit (06396500001; VentanaRoche; Roche Diagnostics GmbH, Mannheim, Germany) was used as the secondary antibody (supplied ready to use). Brown staining was considered positive. Suitable negative and positive controls were used as appropriate [12].

Two pathologists independently evaluated all immunohistochemically stained slides (GL, ST) without knowledge of the clinical information, and the inter-observer variability was below 5%. IHC evaluation was performed by counting 1000 to 1500 tumor cells in a minimum of 10 fields at high magnification per case. Nuclear and cytoplasmic immunoreactivity were evaluated separately. The extent of HDAC expression was calculated by the percentage of positive tumor cells to the total number of tumor cells. Staining intensity was estimated in four categories as follows: 0 (no staining), 1 (mild staining), 2 (moderate staining), and 3 (intense staining). 

### 2.3. Statistical Analysis

Statistical analysis was performed by an MSc biostatistician (GL). In the basic analysis immunohistochemical expression of HDAC-2, HDAC-4, and HDAC-5 was treated as a continuous and/or categorical variable. Associations with clinicopathologic parameters, such as grade, histologic type, and FIGO clinical stage were tested using non-parametric tests, with correction for multiple comparisons, as appropriate, since our data did not have a normal distribution (i.e., Kruskal–Wallis ANOVA, Mann–Whitney U test, and Chi-square test or Fisher’s exact test). Survival analysis was performed using death from the disease as the endpoint. The effect of various parameters on the clinical outcome was assessed in a univariate analysis by plotting survival curves according to the Kaplan–Meier method and comparing groups using the log-rank test, as well as by univariate Cox regression analysis. Multivariate analysis was performed using Cox’s model in order to evaluate the predictive power of each variable independently of the others. A *p*-value of <0.05 was considered statistically significant. A *p*-value of >0.05 but lower than <0.10 was considered of marginal significance. The analysis was performed with the statistical package STATA 11.0/SE for Windows 11.

## 3. Results

### 3.1. Patients’ Characteristics

The patients’ characteristics are shown in Table 1. The median age was 62 years (31–92 years). Eight-six specimens (93.4%) were classified as ovarian adenocarcinomas, whereas six (6.5%) fulfilled the criteria of low malignant potential (LMP) tumors. There were 74 (80.4%) serous carcinomas, three (3.3%) mucinous carcinomas, five (5.4%) endometrioid carcinomas, one (1.1%) clear cell carcinoma, one (1.1%) mixed mucinous–endometrioid carcinoma, two (2.2%) carcinosarcomas, and six (6.5%) serous borderline tumors. Twenty-seven patients (30%) had FIGO stage I, 10 (11.1%) FIGO stage II, 36 (40%) FIGO stage III, and 17 (18.9%) FIGO stage IV. In two cases, the stage was not known. Eighteen patients (21%) had a low-grade (LG) and 68 (79%) had a high-grade (HG) carcinoma. Follow-up information was available for 80 patients, 27 of whom died of their disease within a period of 0.1–77.9 months. The median follow-up time was 25.9 months. Twenty-five patients (39.1%) had lymph node metastasis and 18 (19.6%) had metastatic disease at diagnosis. Sixty-eight patients (73.9%) received platinum-based chemotherapy; sixteen (20.5%) as monotherapy, thirty (38.5%) combined with a Taxane, twenty-one (26.9%) with a Taxane and Bevacizumab, and one patient (1.3%) combined with Doxorubicin.

### 3.2. HDAC-2 Expression and Associations with Clinicopathological Characteristics

HDAC-2 immunopositivity was observed in 94.4% of the examined cases, all cases showing nuclear staining (Figure 1 and Figure 2A,D, Table 2). The majority of the cases displayed an increased immunoexpression (67.8% of the cases with a positive expression in >80% of the tumor population) with a strong staining intensity (61% of the cases). Representative figures of the different staining intensities are shown in Appendix A.

Patients with increased HDAC-2 nuclear immunoexpression were slightly older than those with lower HDAC-2 expression (64 vs. 58 years old, Mann–Whitney U test, *p* = 0.045). Serous histology was mostly associated with higher levels of HDAC-2 expression, which was a correlation of marginal significance (median HDAC-2 expression 80% in non-serous tumors versus 100% in serous tumors, Mann–Whitney U test, *p* = 0.08, Figure 3, Table 3). There was, however, no significant difference in HDAC-2 expression between patients with serous borderline tumors and serous carcinomas (*p* = 0.403). Moreover, there was no statistically significant association between HDAC-2 expression and FIGO stage (Mann–Whitney U test, *p* = 0.169), T-status (Kruskal–Wallis ANOVA, *p* = 0.417), histological grade (Mann–Whitney U test, *p* = 0.531), the presence of metastasis (Mann–Whitney U test, *p* = 0.819), or relapse (Mann–Whitney U test, *p* = 0.967), not only in the whole cohort but also in an analysis restricted to serous tumors [FIGO stage (*p* = 0.692), T-status (*p* = 0.907), histological grade (*p* = 0.255), the presence of metastasis (*p* = 0.866), or relapse (*p* = 0.871) for serous tumors]. Interestingly, the presence of lymph node metastasis was associated with lower levels of HDAC-2 expression, which was an association that was stronger in tumors with serous histology (Mann–Whitney U test, *p* = 0.045, Figure 3) and was of borderline significance in the whole cohort (*p* = 0.081).

### 3.3. HDAC-4 Expression and Associations with Clinicopathological Characteristics

HDAC-4 immunopositivity was cytoplasmic and was observed in 80.4% of the investigated cases (median value of immunoexpression 30%) (Figure 1 and Figure 2B,E, Table 2). The vast majority of the positive cases (77.2%) had a relatively weak staining intensity, whereas only 22.8% displayed a moderate intensity, and none of the cases displayed a strong staining intensity. Representative figures of the different staining intensities are shown in Appendix A.

The associations between cytoplasmic HDAC-4 expression and clinicopathological characteristics are shown in Table 4. There was no significant difference in the levels of HDAC-4 expression between tumors with or without serous histology (Chi-square test, *p* = 0.443). Similarly, the expression in borderline tumors seemed to be comparable to that observed in carcinomas (Chi-square test, *p* = 0.999), and there was no difference among low- or high-grade carcinomas (Chi-square test, *p* = 0.171). FIGO stage IV tumors were more frequently negative for HDAC-4 (35.3%) compared with the tumors with lower FIGO stages (I–III, 16.2%), which was a relationship of marginal significance in the entire cohort (Chi-square test, *p* = 0.07), but was proven to be significant in tumors with serous histology (borderline and carcinomas, N = 80, *p* = 0.05) and especially in serous carcinomas (N = 74, *p* = 0.045). A similar association was observed regarding tumor T-status, especially in the subset of serous carcinomas, where all T1 cases were positive for HDAC-4 compared with 23.6% of the T2/T3 cases (Chi-square test, *p* = 0.043), whereas the respective relationship was of marginal significance in tumors with serous histology (borderline and carcinomas, *p* = 0.078) and failed to attain statistical significance when the analysis was performed on the entire cohort (*p* = 0.353). Likewise, serous carcinomas presenting without metastatic disease at diagnosis displayed a positive immunoreactivity for HDAC-4 more frequently (86.8% vs. 66.7% for cases with metastatic disease, Chi-square test, *p* = 0.05), which was an association of marginal significance when the analysis was performed on the entire cohort (*p* = 0.097). Similarly, tumors without lymph node metastasis were positive for HDAC-4 more frequently compared with tumors with lymph node metastasis, which was an association that was, however, of borderline significance in serous tumors, as well as in the entire cohort (Chi-square test, *p* = 0.069 for the entire cohort, *p* = 0.098 for serous tumors).

### 3.4. HDAC-5 Expression and Associations with Clinicopathological Characteristics

HDAC-5 immunoreactivity in epithelial ovarian carcinomas was mainly cytoplasmic, with some cases displaying additional nuclear immunostaining. Cytoplasmic HDAC-5 expression was observed in 28.1% of the examined cases (25/89) and nuclear only in 10% of the cases (9/89) (Figure 1 and Figure 2C,F, Table 2). The vast majority of the investigated cases did not show any positive expression of HDAC-5 (78.9%). The staining intensity was weak in most of the positive cases (76% of the positive cases). Representative figures of the different staining intensities are shown in Appendix A. There was a positive correlation between nuclear and cytoplasmic HDAC-5 expression (Spearman’s correlation coefficient, rho = 0.53, *p* < 0.001, Table 3).

The associations between cytoplasmic HDAC-5 nuclear or cytoplasmic expression and clinicopathological characteristics are shown in Table 5. There was no significant difference in the levels of HDAC-5 expression (cytoplasmic or nuclear) between tumors with or without serous histology (Chi-square test, *p* = 0.664 for cytoplasmic, *p* = 0.826 for nuclear expression). HDAC-5 expression did not differ between borderline tumors and carcinomas (Chi-square test, *p* = 0.102 for cytoplasmic and *p* = 0.450 for nuclear expression). Moreover, there was no statistically significant association between HDAC-5 expression and FIGO stage (Chi-square test, *p* = 0.348 for cytoplasmic, *p* = 0.785 for nuclear), T-status (Chi-square test, *p* = 0.525 for cytoplasmic, *p* = 0.546 for nuclear), histological grade (Chi-square test, *p* = 0.437 for cytoplasmic, *p* = 0.518 for nuclear), the presence of metastasis (Chi-square test, *p* = 0.227 for cytoplasmic, *p* = 0.473 for nuclear), or relapse (*p* = 0.301 for cytoplasmic, *p* = 0.725 for nuclear). Interestingly, the presence of lymph node metastasis was associated with nuclear HDAC-5 positivity in the entire cohort (Chi-square test, *p* = 0.048).

### 3.5. Associations among HDAC-2, HDAC-4, and HDAC-5

There was a positive correlation between HDAC-4 and HDAC-5 expression (Spearman correlation coefficient, rho = 0.27, *p* = 0.012 for nuclear HDAC-5 and rho = 0.24 *p* = 0.029 for cytoplasmic HDAC-5, Appendix A). These correlations remained significant when the analysis was restricted to serous tumors (Spearman correlation coefficient, rho = 0.28, *p* = 0.014 for nuclear HDAC-5 and rho = 0.26 *p* = 0.026 for cytoplasmic HDAC-5). The associations between HDAC-2 and HDAC-4 or HDAC-5 were not significant. The associations among the investigated HDACs are presented in Table 6.

### 3.6. Survival Analysis

In the univariate survival analysis for overall survival, only HDAC-4 immunopositivity in serous carcinomas seemed to be associated with a better OS, which was a correlation of marginal significance only (log-rank test, *p* = 0.086, Figure 4). The median OS for patients with HDAC-4 expression was 77.9 months compared with 41.53 months for patients without HDAC-4 expression. The respective associations between HDAC-2 and HDAC-5 (nuclear or cytoplasmic) and OS in the entire cohort as well as in serous tumors or serous carcinomas were not significant (log-rank test, for the entire cohort; HDAC-2 *p* = 0.78, nuclear HDAC-5 *p* = 0.324, cytoplasmic HDAC-5 *p* = 0.96, for serous tumors; HDAC-2 *p* = 0.834, nuclear HDAC-5 *p* = 0.509, cytoplasmic HDAC-5 *p* = 0.969). In our cohort tumor T-category (T1 vs. T2 vs. T3, log-rank test, *p* = 0.004), FIGO stage (I vs. II vs. III vs. IV, log-rank test, *p* = 0.004), the presence of relapse (log-rank test, *p* = 0.007) and the presence of residual disease (log-rank test, *p* = 0.035) were associated with an adverse patient prognosis. Table 7 illustrates the results of univariate Cox regression analysis in our cohort.

However, in multivariate survival analysis, none of the investigated HDACs remained significant after adjustment for stage in the entire cohort and in serous carcinomas. In a third Cox regression model, which included all the HDACs and was performed on the advanced-stage tumors (FIGO stage III/IV), none of the investigated HDACs attained statistical significance. The multivariate Cox regression models are presented in Table 8.

## 4. Discussion

Epithelial ovarian cancer (EOC) is one of the leading causes of cancer mortality in women, being characterized by late-stage presentation and poor patient prognosis [11]. Despite the significant progress in the therapeutic approaches for cancer over the past 20 years, which have undoubtedly revolutionized anti-ovarian-cancer therapy, 5-year relative survival rates amount to less than 40% [13]. The first line of treatment encompasses surgery aiming not only at a reduction in tumor volume but also at staging the disease. Cytotoxic chemotherapy with a platinum agent and a Taxane, which can be given before or after surgery, is a mandatory element of treatment [14]. Some cases recur within 6 months after completion of initial chemotherapy and are considered “platinum-resistant” [14]. Chemoresistance is the main challenge for ovarian carcinoma, being responsible for treatment failure and unfavorable clinical outcomes. The dismal prognosis of EOC along with the limitations of the available therapeutic modalities have set the stage for the investigation of the potential use of novel epigenetic therapies, including HDAC inhibitors, either alone or in combination with other therapies.

The word “epigenetics” is derived from the Greek prefix epi (“in annexation, on the top of, all over”), which refers to features that are either “on the extrinsic surface” or “in annexation” to the genetic basis of inheritance. Several epigenetic alterations have been observed in EOC, mainly altered DNA methylation, in terms of pan-hypomethylation of heterochromatin, and regional CpG island methylation [15], with some reports also focusing on the role of histone acetylation [16,17]. In this context, Caslini et al. studied how histone modifications affected the expression of GATA transcription factors on five ovarian cancer cell lines, and more specifically, GATA4 and GATA6 gene silencing was found to correlate with hypoacetylation of histones H3 and H4 [16]. Genome-wide studies have also revealed that genetic alterations affecting the expression of histone-modifying genes are present in ovarian carcinoma [18].

To date, the available information regarding the role of HDACs in EOC is limited to four articles, three of which investigated only members of class I, namely, HDAC-1, -2, and -3 [9,19,20]. These studies report an increased expression of all three class I HDACs (mRNA or immunohistochemical) in ovarian carcinomas, being higher than that observed in normal tissue [9,19,20]. Interestingly, Weichert et al. suggest a lower expression of these class I HDACs in endometrioid ovarian carcinomas compared with serous carcinomas [20]. The fourth study focuses on a member of class IIa, namely, HDAC-4, and presents overexpression of this molecule in EOC, suggesting also a role in the repression of p21 [21]. However, to the best of our knowledge, the expression of HDAC-5 in EOC has not been investigated. Similarly, there is no published information on the potential correlation or interplay among members of different classes of HDACs. 

In accordance with the previous studies investigating HDAC-2 in EOC [9,19,20], in our study, we also observed HDAC-2 immunopositivity in the vast majority of the examined cases. Moreover, HDAC-2 was the most extensively expressed HDAC among the examined HDACs in our cohort. These results are in alignment with previous investigations in uveal melanoma [8,22]. HDAC-4 was also expressed in 80% of the cases, having, however, a relatively weak staining intensity in most of the cases. On the contrary, HDAC-5 was the least expressed HDAC in our cohort, being positive only in 28.1% of the examined cases.

In this study, we observed nuclear HDAC-2 staining, consistent with previous reports in other malignancies as well as in studies on ovarian carcinoma and in keeping with the fact that class I HDACs are reported to be ubiquitously located in the cell nucleus because of a lack of a nuclear export signal [8,20,23]. Most of the published investigations on several human malignancies focus on the role of class I HDACs in the nucleus. Some studies also report cytoplasmic immunoreactivity, the function of which remains unclear [9,12,24]. On the other hand, HDAC-4 exhibited only cytoplasmic and HDAC-5 mainly cytoplasmic immunoreactivity, in keeping with the reported staining patterns of these proteins [8]. The concomitant nuclear and cytoplasmic immunolocalization of HDAC-5 could be attributed to the known capability of class II HDACs of nucleocytoplasmic shuttling in response to certain cellular signals [25].

Importantly, we found that serous tumors (borderline and carcinomas) displayed increased HDAC-2 immunoreactivity; however, this correlation was of marginal significance, probably because of the low number of non-serous tumors included in our investigation. This result is in alignment with the observations reported by Weichert et al. [20], who present an increased expression of HDAC-2 in serous carcinomas compared with endometrioid carcinomas, the latter showing the lowest expression levels among all histological types. 

Interestingly, we observed an inverse association between HDAC-4 expression and lower FIGO stage and tumor T-status. Similarly, HDAC-4 immunopositivity was correlated with the absence of metastatic disease at diagnosis in serous carcinomas. A similar result, although of marginal significance, was observed with the absence of lymph node metastasis. These observations are in contrast to the results of Shen et al., who report increased HDAC-4 expression in ovarian cancer specimens with advanced-stage disease, without however specifying the histological subtype of the investigated cases in their cohort [21]. This inverse association of HDAC-4 observed in our cohort could possibly be an event presenting especially in the subset of serous tumors. According to the literature, HDACs can induce a broad range of cellular and molecular effects through the hyperacetylation of histone and nonhistone substrates and, in this context, regulate and modify a wide range of signaling pathways [10], being therefore able to induce different events in different types of tumors. A similar negative association between HDAC-4 expression and the presence of distant metastasis has also been reported in human pancreatic adenocarcinomas [26]. 

In our study, not only lower HDAC-4 but also lower HDAC-2 expression was associated with the presence of lymph node metastasis. The presence of a similar association between HDAC-2 and -4 and lymph node metastasis is interesting, especially since there was not any significant association between HDAC-2 and HDAC-4. HDAC-2 is the most investigated member of the HDAC family in the literature, showing very frequent associations with parameters correlated with patient prognosis. There are only isolated examples in which HDAC-2 expression levels were not correlated with clinicopathological parameters, such as, for example, in the investigation by Giaginis et al. of pancreatic adenocarcinomas [26]. 

Regarding the associations between HDAC-5 expression and clinicopathological parameters, most of the associations were not proven to be statistically significant, which is probably in accordance with the limited number of positive cases observed in our investigation. Interestingly, the presence of nuclear HDAC-5 immunohistochemical positivity was associated with the presence of lymph node metastasis. Similarly, Peng et al. observed that the suppression of miR-671-5p or promotion of HDAC5 expression encouraged ovarian carcinoma tumor growth in animal models [27]. 

Moreover, we found significant positive associations between the immunohistochemical expression of HDAC-4 and HDAC-5 (nuclear or cytoplasmic), both being class IIa HDACs. This is the first investigation in the literature in which more than one member of the class II family has been investigated. Our result implicates a possible interplay between the two molecules. Further investigations are, however, needed in order to confirm and explain this observation.

Another interesting finding of the present investigation is that the presence of HDAC-4 expression in serous carcinomas connotes a better survival probability. To the best of our knowledge, this appears to be the first report investigating the potential prognostic role of class II HDACs in epithelial ovarian carcinomas. According to a comprehensive review by Weichert, class I HDAC isoforms are mostly expected to be associated with poor patient survival, whereas high expression of class II isoforms seems to predict a better patient outcome [28]. However, the correlation between HDAC-4 and favorable prognosis failed to remain significant in our cohort in the multivariate survival analysis, probably indicating that this association observed in the univariate analysis reflects the respective association between HDAC-4 expression and the FIGO stage.

The present investigation is the first study analyzing concomitantly the expression of members of different classes of the HDAC protein family. It has, however, some limitations that need to be taken into consideration. For example, our cohort encompasses mainly tumors with serous histology, comprising 86.9% of the investigated cases. However, it should be mentioned that the results of the survival analysis in our investigation recapitulate many of the traditional parameters that have been proposed as important determinants of the clinical outcome in epithelial ovarian carcinomas, namely, FIGO stage and tumor T-category as well as the presence of relapse or residual disease after surgery, supporting the validity of our statistical analysis and denoting that our cohort is representative.

In the last decade, a variety of compounds that can block the deacetylase activity of HDACs have been recognized, and synthetic or natural molecules targeting class I, II, and IV HDACs have been developed [29]. Moreover, preclinical studies have found that HDAC inhibitors (HDACis) are able to inhibit ovarian cancer cell growth in vitro and in vivo by inhibiting the cell cycle and inducing mitotic defects through histone-mediated and histone-independent interactions [30]. Recently, numerous clinical trials investigated the role of HDACis alone or in combination with other drugs in patients with epithelial ovarian cancer, showing encouraging anti-tumor effects [29], especially when combined with other chemotherapeutics. This combination seems to show chemosensitizing or synergistic antitumor efficacy, which may be due to their ability to overcome particular mechanisms associated with drug resistance [31]. However, additional studies are needed to determine the efficacy of these therapies.

## 5. Conclusions

The present investigation is the first study to simultaneously investigate the immunohistochemical expression of HDAC members of different classes and show clearly that different members of the HDAC family are differentially expressed in EOC, HDAC-2 being the most frequently and HDAC-5 the least frequently expressed isoform among the investigated HDACs. Additionally, significant positive associations between HDAC-4 and HDAC-5 expression levels were observed, implicating a potential interplay among members of class II. HDAC-4 expression was mostly correlated with clinicopathological parameters, being associated with lower FIGO stage and tumor T-category as well as the absence of lymph node metastasis or distant metastasis. These findings provide evidence for the potential role of HDACs in the biological mechanisms governing epithelial ovarian cancer evolution and progression.

## Figures and Tables

**Figure 1 biomedicines-12-00947-f001:**
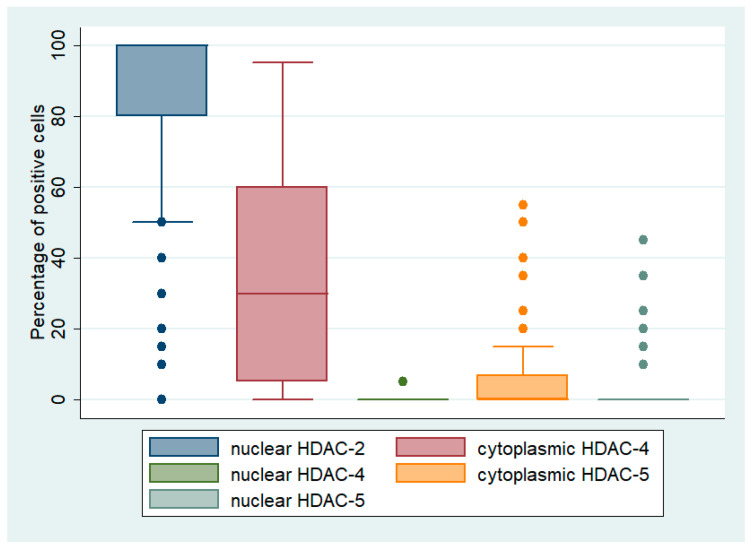
Box-plot graph showing the expression levels in the investigated samples.

**Figure 2 biomedicines-12-00947-f002:**
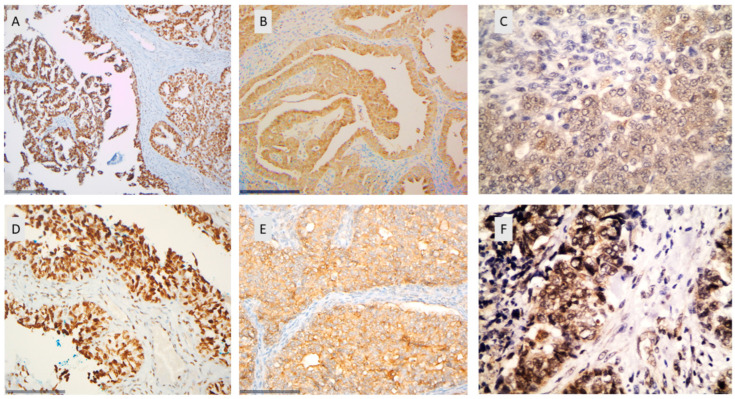
Immunohistochemical expression of HDAC-2 (**A**,**D**), HDAC-4 (**B**,**E**), and HDAC-5 (**C**,**F**) in epithelial ovarian carcinomas. (**A**) A high-grade serous ovarian carcinoma with nuclear HDAC-2 expression (×200). (**B**) A high-grade serous ovarian carcinoma with cytoplasmic HDAC-4 expression (×200). (**C**) A high-grade serous ovarian carcinoma with cytoplasmic HDAC-5 expression (×400). (**D**) A high-grade serous ovarian carcinoma with nuclear HDAC-2 expression (×400). (**E**) A high-grade serous ovarian carcinoma with cytoplasmic HDAC-4 expression (×400). (**F**) A high-grade serous ovarian carcinoma with nuclear and cytoplasmic HDAC-4 expression (×400).

**Figure 3 biomedicines-12-00947-f003:**
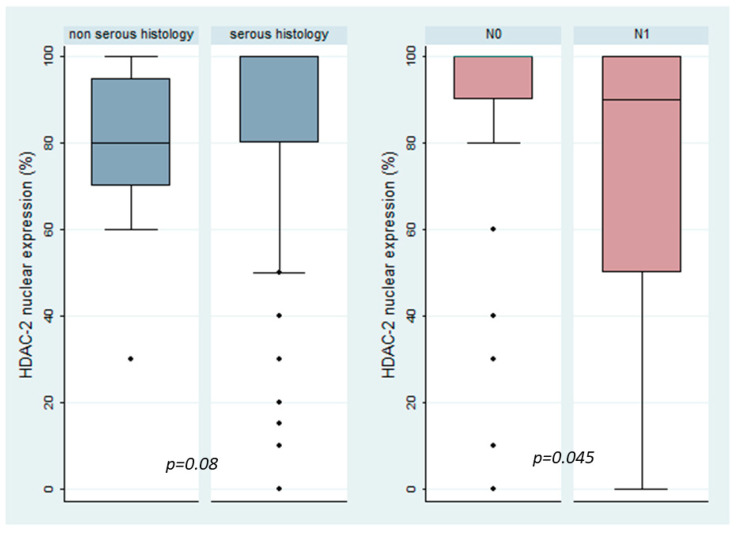
Associations between HDAC-2 nuclear expression and serous histology (**first panel**) and with the presence of lymph node metastasis at the time of diagnosis in serous carcinomas (**second panel**).

**Figure 4 biomedicines-12-00947-f004:**
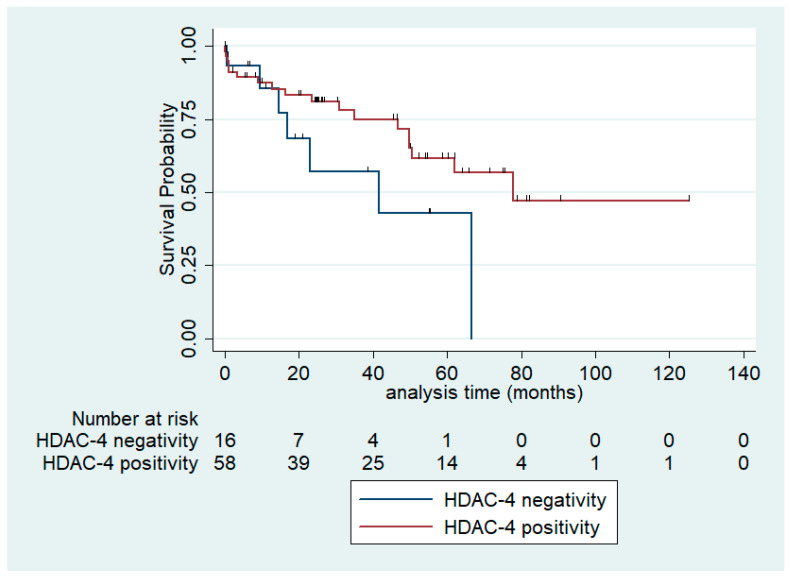
Kaplan–Meier survival curves according to cytoplasmic HDAC-4 positivity in serous carcinomas.

**Table 1 biomedicines-12-00947-t001:** Patients’ characteristics.

Patients’ Characteristics	Median Value	Min–Max
**Age (in years)**	62	31–92
	**Number of patients**	**Percentage**
**Histological subtype**		
Serous carcinoma	74	80.4%
Mucinous carcinoma	3	3.3%
Endometrioid carcinoma	5	5.4%
Clear cell carcinoma	1	1.1%
Mixed mucinous–endometrioid carcinoma	1	1.1%
Carcinosarcoma	2	2.2%
Serous borderline tumor	6	6.5%
**FIGO stage**		
I	27	30%
II	10	11.1%
III	36	40%
IV	17	18.9%
NA	2	
**Tumor grade**		
Low grade	18	21%
High grade	68	79%
**T-status**		
T1	29	32%
T2	11	12%
T3	50	56%
NA *	2	
**N-status**		
N0	39	60.9%
N1	25	39.1%
**Metastasis**		
Metastatic cancer	18	19.6%
Non-metastatic cancer	74	80.4%
**Residual disease**		
R0	25	29.1%
R1/R2	10	11.6%
Unknown	51	59.3%
**Event**		
Death from disease	27/80 (follow-up: 0.1–77.9 months, median OS 16.9 months)	28%
Censored	53/80 (follow-up: 0.3–125.2 months)	71%
**Relapse**		
Present	36	58.1%
Absent	26	
NA	30	
**Adjuvant chemotherapy**		
Platinum	16	20.5%
Platinum, Taxane	30	38.5%
Platinum, Taxane, Bevacizumab	21	26.9%
Platinum, Doxorubicin	1	1.3%
None	10	12.8%
NA	14	

* NA: not available.

**Table 2 biomedicines-12-00947-t002:** Expression of the investigated HDACs in epithelial ovarian carcinomas.

	Positivity Rate *	Median	Min–Max
Nuclear HDAC-2	94.4%	100	0–100
Cytoplasmic HDAC-4	80.4%	30	0–95
Cytoplasmic HDAC-5	28.1%	0	0–55
Nuclear HDAC-5	10%	0	0–45

* Number of positive cases throughout the cohort.

**Table 3 biomedicines-12-00947-t003:** Associations between nuclear HDAC-2 expression and clinicopathological characteristics. Results of the Mann–Whitney U test.

Variable	Nuclear HDAC-2	
	**Whole Cohort**	**Serous Tumors**
	(median, min–max)	(median, min–max)
**Histologic type**		
Serous histology	100 (0–100)	
Non-serous histology	80 (30–100)	
	*p* = 0.080	
**Grade**		
Low	87.5 (10–100)	100 (10–100)
High	100 (0–100)	100 (0–100)
	*p* = 0.531	*p* = 0.255
**FIGO stage**		
I/II	90 (0–100)	100 (0–100)
III/IV	100 (0–100)	100 (0–100)
	*p* = 0.169	*p* = 0.692
**T-status**		
T1	100 (0–100)	100 (0–100)
T2	85 (0–100)	100 (0–100)
T3	100 (0–100)	100 (0–100)
	*p* = 0.417	*p* = 0.907
**N-status**		
N0	100 (0–100)	100 (0–100)
N1	90 (0–100)	90 (0–100)
	*p* = 0.081	*p* = 0.045
**Metastasis**		
Metastatic cancer	97.5 (0–100)	97.5 (0–100)
Non-metastatic cancer	100 (0–100)	100 (0–100)
	*p* = 0.819	*p* = 0.866
**Relapse**		
Present	100 (0–100)	100 (0–100)
Absent	100 (0–100)	100 (0–100)
	*p* = 0.967	*p* = 0.871

**Table 4 biomedicines-12-00947-t004:** Associations between cytoplasmic HDAC-4 expression and clinicopathological characteristics. Results of the Chi-square test.

Variable	Cytoplasmic HDAC-4			
	**Whole Cohort**		**Serous Tumors**	
	negative	positive	negative	positive
	N (%)	N (%)	N (%)	N (%)
**Histologic type**				
Serous histology	3 (27.3%)	8 (72.7%)		
Non-serous histology	14 (18.4%)	62 (81.6%)		
	*p* = 0.443			
**Grade**				
Low	1 (5.9%)	16 (94.1%)	0 (0%)	9 (100%)
High	15 (23.1%)	50 (76.9%)	13 (20.9%)	49 (79.1%)
	*p* = 0.171		*p* = 0.195	
**FIGO stage**				
I/II/III	11 (16.2%)	57 (83.8%)	8 (14%)	49 (86%)
IV	6 (35.3%)	11 (64.7%)	6 (35.3%)	11 (64.7%)
	*p* = 0.070		*p* = 0.050	
**T-status**				
T1	3 (11.5%)	23 (88.5%)	1 (5.3%)	18 (94.7%)
T2/T3	11 (22.9%)	37 (77.1%)	13 (23.6%)	42 (76.4%)
	*p* = 0.353		*p* = 0.078	
**N-status**				
N0	3 (8.1%)	34(91.9%)	2 (6.45%)	29 (93.5%)
N1	6 (25%)	18 (75%)	5 (21.7%)	18 (78.3%)
	*p* = 0.069		*p* = 0.098	
**Metastasis**				
Metastatic cancer	6 (33.3%)	12 (66.7%)	6 (33.3%)	12 (66.7%)
Non-metastatic cancer	11 (15.9%)	58 (84.1%)	8 (13.8%)	50 (86.2%)
	*p* = 0.097		*p* = 0.062	
**Relapse**				
Present	8 (23.5%)	26 (76.5%)	7 (21.2%)	26 (78.8%)
Absent	3 (13%)	20 (87%)	3 (13.6%)	19 (86.4%)
	*p* = 0.325		*p* = 0.475	

**Table 5 biomedicines-12-00947-t005:** Associations between nuclear and cytoplasmic HDAC-5 expression and clinicopathological characteristics. Results of the Chi-square test.

Variable	HDAC-5 Expression			
	**Cytoplasmic**		**Nuclear**	
	negative	positive	negative	positive
	N (%)	N (%)	N (%)	N (%)
**Histologic type**				
Serous histology	8 (66.7%)	4 (33.4%)	11 (91.7%)	1 (8.3%)
Non-serous histology	56 (72.7%)	21 (27.3%)	69 (89.6%)	8 (10.4%)
	*p* = 0.664		*p* = 0.826	
**Grade**				
Low	12 (66.7%)	3 (33.3%)	17 (94.4%)	1 (5.6%)
High	50 (75.8%)	16 (24.2%)	59 (89.4%)	7 (10.6%)
	*p* = 0.437		*p* = 0.518	
**FIGO stage**				
I/II	23 (65.7%)	12 (34.3%)	31 (88.6%)	4 (11.4%)
III/IV	39 (75%)	13 (25%)	47 (90.4%)	5 (9.6%)
	*p* = 0.348		*p* = 0.785	
**T-status**				
T1	18 (66.7%)	9 (33.3%)	25 (92.6%)	2 (7.4%)
T2/T3	44 (73.3%)	16 (26.7%)	53 (88.3%)	7 (11.7%)
	*p* = 0.525		*p* = 0.546	
**N-status**				
N0	30 (78.9%)	8 (21.1%)	37 (97.4%)	1 (2.6%)
N1	16 (66.7%)	8 (33.3%)	20 (83.3%)	4 (16.7%)
	*p* = 0.282		*p* = 0.048	
**Metastasis**				
Metastatic cancer	15 (83.3%)	3 (16.7%)	17 (94.4%)	1 (5.6%)
Non-metastatic cancer	49 (69%)	22 (31%)	63 (88.7%)	8 (11.3%)
	*p* = 0.227		*p* = 0.473	
**Relapse**				
Present	27 (75%)	9 (25%)	32 (88.9%)	4 (11.1%)
Absent	15 (62.5%)	9 (37.5%)	22 (91.7%)	2 (8.3%)
	*p* = 0.301		*p* = 0.725	

**Table 6 biomedicines-12-00947-t006:** Associations among the investigated HDACs in epithelial ovarian carcinomas. Star denotes a statistically significant result.

	Nuclear HDAC-2	Cytoplasmic HDAC-4	Cytoplasmic HDAC-5
Cytoplasmic HDAC-4	rho = −0.09 *p* = 0.390		
Cytoplasmic HDAC-5	rho = −0.03 *p* = 0.781	rho = 0.24 *p* = 0.029 *	
Nuclear HDAC-5	rho = −0.03 *p* = 0.792	rho = 0.27 *p* = 0.012 *	rho = 0.5274 *p* < 0.001 *

**Table 7 biomedicines-12-00947-t007:** Univariate Cox regression analysis of our cohort.

Parameter	Entire Cohort			Serous Carcinomas		
	*p*-Value	Hazard Ratio	95% CI	*p*-Value	Hazard Ratio	95% CI
FIGO stage	0.002	2.05	1.31–3.22	0.001	2.5	1.46–4.31
T-status	0.014	2.13	1.16–3.89	0.014	2.66	1.22–5.80
Histological grade	0.367	1.73	0.52–5.76	0.043	1.81	1.02–2.05
Serous histology	0.060	0.31	0.08–1.12	-	-	-
Presence of metastasis	0.116	1.98	0.82–4.77	0.073	2.28	0.92–5.62
Presence of lymph node metastasis	0.112	2.09	0.84–5.22	0.183	1.96	0.72–5.28
Presence of residual disease	0.052	4.17	0.98–17.62	0.041	5.55	1.07–28.71
Presence of relapse	0.019	5.77	1.34–24.84	0.029	5.13	1.18–22.29
HDAC-2 positivity	0.789	1.31	0.17–9.83	0.835	1.23	0.16–9.32
HDAC-4 positivity	0.113	0.48	0.20–1.18	0.086	0.50	0.19–1.30
HDAC-5 cytoplasmic positivity	0.967	0.98	0.42–2.34	0.970	0.98	0.38–2.49
HDAC-5 nuclear positivity	0.330	1.70	0.58–5.01	0.512	1.51	0.44–5.12

**Table 8 biomedicines-12-00947-t008:** Multivariate Cox regression analysis of the entire cohort (Model A), of serous tumors (Model B), and of advanced-stage (FIGO III/IV) tumors (Model C).

Model	Parameter	Hazard Ratio	SE	*p*-Value	95% CI	
A (N = 73)	Cytoplasmic HDAC-4 positivity	0.58	0.28	0.255	0.23	1.48
	Nuclear HDAC-2 positivity	1.31	1.37	0.797	0.17	10.20
	Nuclear HDAC-5 positivity	1.85	1.73	0.509	0.30	11.56
	Cytoplasmic HDAC-5 positivity	0.65	0.41	0.493	0.18	2.26
	FIGO	2.06	0.54	0.006	1.23	3.45
B (N = 66)	Cytoplasmic HDAC-4 positivity	0.64	0.33	0.381	0.23	1.75
	Nuclear HDAC-2 positivity	1.20	1.27	0.861	0.15	9.56
	Nuclear HDAC-5 positivity	1.92	1.81	0.487	0.30	12.16
	Cytoplasmic HDAC-5 positivity	0.69	0.45	0.566	0.19	2.48
	FIGO	2.46	0.76	0.003	1.35	4.50
C (N = 46)	Cytoplasmic HDAC-4 positivity	0.54	0.28	0.232	0.20	1.48
	Nuclear HDAC-2 positivity	1.21	1.30	0.858	0.15	9.93
	Nuclear HDAC-5 positivity	2.12	1.99	0.423	0.34	13.32
	Cytoplasmic HDAC-5 positivity	0.75	0.49	0.662	0.21	2.69

## Data Availability

The data presented in this study are available on request from the corresponding author.

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
