# Peer review of "Clinical Significance of the Immunohistochemical Expression of Histone Deacetylases (HDACs)-2, -4, and -5 in Ovarian Adenocarcinomas"

_biomedicines, 2024, doi:10.3390/biomedicines12050947_

Round 1
Reviewer 1 Report
Comments and Suggestions for Authors
The results presented in the manuscript have been previously published, although the Authors declared that the results have not been published elsewhere previously (https://www.preprints.org/manuscript/202403.1519/v1)
Additionally, in my opinion, if the article were to be published in the journal Biomedicinces, the diagrams should be corrected to make them more readable. Moreover, in the discussion, the Authors once again describe in detail the previously described results. The discussion should be written more in terms of the clinical application of the results obtained.
Comments on the Quality of English LanguageModerate editing of English language required.
Author Response
Dear sir/madam,
Thank you for your comments.
- The results presented in the manuscript have been previously published, although the Authors declared that the results have not been published elsewhere previously (https://www.preprints.org/manuscript/202403.1519/v1)
The results of this manuscript have not been previously published. Our article has been recommended upon submission to Biomedicines from the editorial office of Biomedicines (Email on the 25th March) to the Preprints.org platform. This is a platform to protect our idea from being stolen and is not considered prior publication.
- Additionally, in my opinion, if the article were to be published in the journal Biomedicinces, the diagrams should be corrected to make them more readable. Moreover, in the discussion, the Authors once again describe in detail the previously described results. The discussion should be written more in terms of the clinical application of the results obtained.
All the figures included in the first version of our manuscript have now being modified (according also to the recommendations of the 2nd Reviewer). Moreover, the Discussion section has been reduced into 1571 words, only 273 of those referring to the results of our investigation. The rest of the text in this sections focuses on the possible explanation of our results or their probable clinical application.
- Moderate editing of English language required.
Our manuscript has been thoroughly reviewed by a native English speaker.
Reviewer 2 Report
Comments and Suggestions for Authors
Dear Authors,
I read your work with great interest. It is a small and simple study but I believe the work can be published after some improvements.
The introduction and the materials and methods section should be rephrased. There are too many sections identical with other works from published literature.
Please report nominal variables as N(%). You sometimes use N, sometimes %. Inconstant data reporting is unpleasant to read.
Please choose either , or . for number decimals.
The rest of my comments can be found in the pdf
Should the results change after applying these suggestions, make sure to adjust the discussion accordingly.

Comments on the Quality of English LanguageSome syntax errors to be corrected. I tried marking all of them
Author Response
Thank you for your valuable suggestions focusing on the improvement of our manuscript.
- The introduction and the materials and methods section should be rephrased. There are too many sections identical with other works from published literature.
The introduction and the materials and methods sections (especially the parts of the manuscript, which were shown to have a high duplication rate, according to the official check by the editorial office) have been rephrased.
- Please report nominal variables as N(%). You sometimes use N, sometimes %. Inconstant data reporting is unpleasant to read.
All nominal variables are now reported as N(%).
- Please choose either , or . for number decimals.
We now use the same punctuation for number decimals.
- The rest of my comments can be found in the pdf
Regarding your comments throughout the pdf of the article:
- The same punctuation (namely “.”) is used.
- We are now more specific with the term “differential expression”.
- Space is added after the word histones.
- References are added throughout the second paragraph of Introduction section.
- Inclusion and exclusion criteria are now included.
- Details about the immunohistochemical technique (Autostainer, detection Kit etc.) are now included.
- Secondary sites were not analyzed, and this is now explained in the exclusion criteria.
- Non-parametric tests were used, since our data did not have a normal distribution.
- The sentence “Interventionary studies involving animals or humans, and other studies that require ethical approval, must list the authority that provided approval and the corresponding ethical approval code” has been deleted.
- Information regarding chemotherapy has been added.
- Number and percentages has been added in the whole paragraph “Patients’ characteristics”
- The sentence “For two cases there was not any information regarding tumor stage” has been rephrased.
- Median follow up time has been added.
- Table 1: NA cases (FIGO stage) have been added.
- Residual disease in Table 1 has been now explained using R.
- Table 1: median OS has been added.
- Table 1: instead of censored we have added absence of relapse and NA
- Table 1: median relapse time and 95% CI interval has not been added, since we did not have information for time to relapse. We only had information for the presence of relapse.
- Tables with the Associations of HDAC-2, -4 and -5 with Clinicopathological Characteristics have been added (new Tables 3, 4, 5).
- Exact p-values have been also added in the text.
- The phrase not any significant has been replaced with no significant.
- Table 2: we now explain the term positivity rate.
- Table 2: we replaced the word range with Min-Max.
- A box-plot graph showing the expression levels in the samples has been added (Figure 1 in the revised version of the manuscript).
- “Advanced FIGO stage tumors” has been replaced with “FIGO stage tumors IV”.
- Number of borderline serous tumors and carcinomas has been added.
- Inappropriate split of the text has been taken into consideration.
- New Figure 2 (previous Figure 1) has been rotated. A panel with a (x400) magnification for each marker has been added.
- A supplementary picture with different staining intensities has been added.
- FIGO stage tumors IV and the absence of metastatic disease were correlated with absence of HDAC-4 expression. I suppose the word “without” before metastatic disease was not taken into consideration due to the previously mentioned split of the text.
- New Figure 3 (old Figure 2): p-values have been added and the y-axis is now the same.
- We decided to make graphs for HDAC-2 only for the comparisons which were proven to be significant and describe the remaining comparisons in the text, as well as in tables.
- A supplementary graph illustrating the associations between nuclear and cytoplasmic HDAC-5 with HDAC-4 is added.
- The associations between HDAC-2, -4 and -5 in the subset of serous tumors is now clearly presented.
- Progression-free survival has not been added since this information was not available in our cohort.
- New Table 7 (previous Table 4): we added univariate Cox regression analysis with HR, and 95% CI.
- We also added in detail the results of log rank test for HDAC-2 as well as HDAC-5 positivity in the text. Median OS for HDAC-4 positivity and negativity has also been included.
- The results of multivariate survival analysis are now presented in Table 8. We adjusted three different models. In the first we added all investigated molecules as well as FIGO stage. As suggested T, N, and metastasis as well as presence of residual disease were not included since they are correlated with FIGO. In the second model the analysis was restricted to serous tumors. In the third model we performed an analysis only in advanced stage tumors as suggested by the reviewer.
- Figure 4 (previous Figure 3): In the revised version of the figure we used colored lines and we included censored as well as at risk table.
- Discussion section: reference has been added.
- The sentence “, HDAC molecules seem to have a different role in serous tumors compared with other histological types” has been omitted.
- The last sentence in the paragraph of conclusions has also been omitted.
- Some syntax errors to be corrected. I tried marking all of them.
Thank you for your help. Our manuscript has also been thoroughly reviewed by a native English speaker.
Round 2
Reviewer 1 Report
Comments and Suggestions for Authors
Accepted.
Author Response
Thank you for your interest in your manuscript.
I assume that there are no suggestions for modifications.
Reviewer 2 Report
Comments and Suggestions for Authors
dear Authors,
Thank you for taking the time to go through my comments.
My last suggestions: please make all graphs using the same style for consistency.
table 8: you did not mention model C
Best of luck,
Author Response
Dear sir/madam,
Thank you for your interest in our manuscript.
We modified our article according to your suggestions.
- please make all graphs using the same style for consistency.
We modified the graph in Figure 3, so that all graphs have the same style for consistency.
- table 8: you did not mention model C.
We now mention model C in the Figure legend.